# Characterization of Extruded Sorghum-Soy Blends to Develop Pre-Cooked and Nutritionally Dense Fortified Blended Foods

**DOI:** 10.3390/foods14050779

**Published:** 2025-02-25

**Authors:** Michael Joseph, Qingbin Guo, Brian Lindshield, Akinbode A. Adedeji, Sajid Alavi

**Affiliations:** 1Prestage Department of Poultry Science, North Carolina State University, Raleigh, NC 27695, USA; 2Department of Grain Science and Industry, Kansas State University, 201 Shellenberger Hall, Manhattan, KS 66502, USA; guoqingbin008322@gmail.com (Q.G.); salavi@ksu.edu (S.A.); 3Department of Food, Nutrition, Dietetics and Health, 245 Justin Hall, Manhattan, KS 66506, USA; blindsh@k-state.edu; 4Department of Biosystems and Agricultural Engineering, University of Kentucky, 128 C E Barnhart Building, Lexington, KY 40546, USA

**Keywords:** sorghum, soy, fortified blended foods, extrusion, anti-nutritional factors, water solubility index, water absorption index, gelatinization, digestibility

## Abstract

Food aid commodities are essential food items in global food aid programming. Some are primarily made from an extrusion of corn and soybeans. However, there are concerns about the genetically modified organisms (GMOs) of some of these grains. Hence, there is a need for alternatives to grains, like sorghum, which is not GMO. It is critical to ensure that products from this new ingredient meet the quality requirements, hence the need to profile them. An expanded formulation sorghum-soy blend (SSB), obtained from extrusion cooking, was ground using a hammer mill and analyzed for changes in properties that were affected by the transformation of starch and protein during processing. Macro- and micro-nutrients were added to these milled blends to prepare fortified blended foods (FBFs) that could meet the recommendations of Food Aid Quality Review (FAQR) report on energy, protein, and micronutrient content. The water absorption index (WAI) ranged from 2.82 to 5.90 g/g, the water solubility index (WSI) ranged from 6.22 to 18.50%, and the blends were affected by the formulation—whole/decorticated sorghum and different levels of fat. Extrusion processing caused starch gelatinization in the range of 90.69–96.26%. The pasting properties indicated that whole grain blends of SSB had lower peak time and higher final viscosity when compared to decorticated sorghum blends. The Bostwick flow rate of cooked porridges with 20% solids was within the recommended range of 9–21 cm/min. Starch digestibility significantly increased after extrusion, with a 149.65% increase in rapidly digestible starch (RDS). The protein digestibility did not vary significantly when subjected to extrusion and wet cooking. There was a significant reduction in anti-nutritional factors in the extruded binary blends of SSB when compared to respective raw blends: phytic acid was reduced by 25.33%, tannins were not found, and trypsin inhibitors were reduced by 19.50%. Thus, the extrusion processing of SSB with the subsequent addition of macro- and micro-ingredients was effective in producing FBFs with high nutritive value, comparable to FBF made from traditional ingredients.

## 1. Introduction

Hunger is a pervasive problem in emerging economies, undermining people’s health, productivity, and often their very survival [1]. According to the report in “The State of Food Insecurity in the World 2024” by the United Nations Food and Agriculture Organization, an estimated 713 to 757 million people of the 8.3 billion people in the world, or 8.9 to 9.4%, were suffering from chronic undernourishment as of 2023. The majority of these people live in developing countries. It is estimated that 20.4% of Africans, 8.1% of Asians, 6.2% of Latin Americans, and 7.3% of Oceanians were facing hunger as of 2023. Since 2014, the prevalence of undernourishment of the global population has risen from 7.3% to 9.4% globally and with similar margins for the developing countries. [2]. This increasing trend in hunger and lack of access to basic nutrition, especially in the sub-Saharan region of Africa, with the highest prevalence amongst any region of the world, calls for continued efforts to stem the trend.

Population growth, poverty, conflicts, social exclusion, governance, trade policies, inequality, and natural disasters are some of the causes of food insecurity in the world [3]. Therefore, the importance of food aid in addressing certain food insecurity issues becomes paramount. Food aid has been used as an instrument to offset food shortages in low-income countries, where fluctuations in domestic food production threaten food security [4]. Fortified blended foods (FBFs) are not only ‘ready-made’ in the sense that they are nutritionally rich in their physical form with easy preparation methods, but usually they are dispensed through a standardized regime, involving registration, anthropometric measurement, and cooking and hygiene training [5,6,7]. The United States is the leading contributor to international food aid, with an average supply of 56% of the annual total food aid donated by members of the Food Aid Committee of the International Grains Council since 1995 [8]. Food aid programs in the U.S. are administered by the United States Agency for International Development (USAID) and the United States Department of Agriculture (USDA), either as part of a bilateral program or through the United Nation’s World Food Program. This food aid is distributed by the U.S. under four authorities: (1) the Food for Peace Act (FFPA, also known as P.L. 480); (2) the Section 216(b) program (inactive since 2007); (3) the Food for Progress Act of 1985; and (4) the McGovern–Dole International Food for Education and Child Nutrition Program [9]. In 1996, the law that governs U.S.-supported FBF programming was amended to permit the enrichment and fortification of commodities to improve their nutritional quality and include high protein blends of U.S. foods for malnourished infants, children, pregnant women, and lactating mothers. Foods donated under P.L. 480 today include whole commodities, processed foods, fortified processed foods, and blended food supplements.

The fortified foods routinely combine cereals and soybean meal products to increase the quality and quantity of proteins. Wheat, corn, sorghum, rice, and soy are often processed, fortified, or enriched into products such as corn-soy blends (CSBs), wheat-soy blends (WSBs), fortified wheat flour, fortified cornmeal, and vitamin A-fortified vegetable oil. CSB is often used to treat moderate malnutrition and micronutrient deficiencies in underweight children. CSB is the most commonly programmed specialized product in supplementary feeding programs [10]. In the early 1970s, there was a shortage of nonfat dried milk, which was used regularly in corn-soy milk (CSM) in food aid programs. This led to the reformulation to CSB. The ineffectiveness of CSB, which is classified as ready-to-use supplementary food, in addressing moderate acute malnutrition, is due to the inadequate compositional profile of energy density, micronutrients, and lipids. CSB has developed expeditiously with different variants, such as CSB 10, 11, 12, and 13 and CSB Plus between 2005 and 2011, with enhancements to address the earlier shortcomings. In Title II programming, the fortified blends and fortified vegetable oils account for 44% of the commodity cost, though this fortification constitutes only 20% of the volume [11]. The Food Aid Quality Review (FAQR) report on energy, protein, and micronutrient content recommends the use of other cereals, namely sorghum, millet, and rice, instead of traditional cereals like corn and wheat, in the production of FBF. The major advantage of using non-GMO crops like sorghum is that it would be well received by governments of target nations, like countries in Africa. Additionally, crops like sorghum, millet, cowpea, etc., are plants that require less water and can thrive in drought like conditions [12], which would add to the promotion of sustainable agriculture. Corn experiences pre-harvest insect damage due to shallower rooting systems as well as wet post-harvest and storage practices in underdeveloped economies, leading to increased risk of aflatoxin contamination [13].

Ideally, the ingredients for low-cost weaning formulations must be derived from dietary staples from the region of interest that are affordable to the section of the target population and readily available in sufficient quantity [14]. Therefore, in this study, sorghum and soy were combined in a ratio (75:25) that would provide the recommended protein (≥18 g) and energy (>350 kcal) when added with other ingredients in the FBFs [11]. Grain sorghum contains phytochemicals such as phenolic compounds, plant sterols, and policosanols, which are rich in antioxidants and impact human diets significantly by lowering cholesterol and promoting cardiovascular health. The condensed tannins present in phytochemicals seem to have powerful anti-carcinogenic and anti-diabetic in vitro activity [15]. However, the bioavailability of tannins (procyanidins and catechins) is questionable due to their larger molecular size and tendency to bind food molecules into insoluble complexes, making human digestion difficult. Another anti-nutritional factor is the phytic acid (inositol hexaphosphate (IP6)) in cereal-legume-based complimentary foods, which inhibits iron absorption from porridges, leading to a high prevalence of iron deficiency in infants [16]. Studies show that dephytinization through processing in low tannin sorghum increased iron absorption by twofold in sorghum reconstituted with water [17]. Heat treatment during processing has also shown encouraging results in lowering phytates, which increase iron solubility by forming iron complexes in naturally occurring plant phytates [18]. Extrusion has been successfully used in the production of low-cost cereal-based weaning foods [19,20]. The processing of cereal-legume blends by extrusion has been used to completely gelatinize starch at 150–170 °C barrel temperatures with a moisture range of 16–22% [21] as well as to denature proteins, enabling a nutritious precooked blended product. When processed under ideal conditions using extrusion technology, catechins and procyanidins in tannins showed improved bioavailability of up to 50% in diets [22] and the possible breakdown of high molecular weight polymers of procyanidins, making it easier for human absorption and thereby improving the nutraceutical value of sorghum [23]. Extrusion heat treatment and shear forces further inactivate trypsin inhibitors by 90% in extrudates [24], thus retaining most of the chemically available lysine in soy flour when extruded at 100 to 115 °C with 12 to 18% in-barrel moisture [25]. The idea behind researching other grains such as sorghum for use in FBF production is to utilize viable nutrient sources in order to bring down production costs, increase acceptance, and implement new nutrient recommendations listed in the FAQR report.

The sorghum-soy blend (SSB), as it will be called, is a precooked blend using extrusion and micronutrient fortification; it is a possible solution for wasted infants and those with stunted growth in areas where FBFs are served. The objective of this study was to develop an SSB that conforms to the most recent recommendations from FAQR using extrusion processing and to evaluate the quality attributes of the extruded SSB in comparison with traditional CSBs. Specific processing-to-product relationships were established, which included post-milling particle size, viscosity profiles, and the effects on Bostwick flow rate, which is related to gruel consistency for infant consumption, starch and protein digestibility, and anti-nutritional factors.

Some of the FAQR key recommendations to better supplement nutritional targets along with optimal breastfeeding in infant feeding practices were to

(1)Increase the quantity and quality of protein by the addition of animal protein, namely WPC-80 (whey protein concentrate);(2)Increase the caloric content of FBFs by adding vegetable oil to post-extruded binary blend;(3)Upgrade the micronutrient composition.

In order to achieve these recommended targets, the study objectives were to investigate (a) the physicochemical and functional properties of the binary blends prepared by using commercial flours or milled flours and their respective FBFs and (b) the influence of the formulation on the viscosity of the porridge.

## 2. Materials and Methods

### 2.1. Materials

The following material components were used: (1) extruded and milled binary blends of SSB (consisting of 75% sorghum and 25% soy). Two white varieties of sorghum (V1 and V2) were purchased from Nu Life Market, Scott City, KS, USA. These sorghum grains were milled as described in [26]. The milled sorghum was either whole (denoted by WS) or decorticated (denoted by the first S in the formulation). The commercially milled sorghum of the white variety, either as whole flour or decorticated flour, was also purchased from the same vendor. These formulations have the suffix ‘com’. The soybean flour, which is denoted by the second S in the formulation, was purchased from American Natural Soy, Cherokee, IA, USA. The soybean flour was low fat (denoted by S), medium fat (denoted by S′), or full fat (denoted by S″). Another group of soybean flour was called aspirated soy (denoted by the suffix ‘aspi’), which was produced by aspirating the hulls from the full-fat soy flour; (2) sugar (C&H brand granulated white cane sugar, purchased from a local store, Manhattan, KS, USA); (3) whey protein concentrate WPC80 (Davisco Foods International, Inc., Eden Prairie, MN, USA); and (4) vitamins and minerals (Research Products Company, Salina, KS, USA) and non-GMO soybean oil (Zeeland Farm Services, Inc., Zeeland, MI, USA).

### 2.2. Extrusion Processing

The binary formulation of sorghum and soy was extruded on a pilot-scale single screw Wenger X-20 (Wenger Manufacturing Inc., Sabetha, KS, USA). The extruder screw profile is shown in Figure 1. The L/D ratio was 8.11. The raw material feed rate was 166 kg/h, and the calculated in-barrel moisture content was 24%. The extruder screw speed ranged from 500 to 550 rpm. The extruder had three temperature zones that were set at increasing temperatures from Zone 1 to Zone 3 as 60 °C, 70 °C, and 90 °C, respectively. These temperature settings caused the final die temperature to be approximately 140 °C. The die had a single circular opening of 4.1 mm, and the extrudates were dried to a moisture content below 5% using a Wenger Double Pass Dryer/Cooler (Series 4800, Wenger Manufacturing Inc., Sabetha, KS, USA) being operated at 104 °C. The dried extrudates were cooled for 10 min by passing the extrudates through a cooler blowing ambient air provided at the end of the dryer. All the process conditions were set based on the prior multiple runs to optimize the processing conditions to extrude the binary blends of SSB containing different proximate compositions. The aim of the optimization/trial runs was to determine the minimum surging occurrence in the extruder by manipulating the steam lock size and sequence as shown in Figure 1. Also, different formulations were run to ascertain the best screw profile. No specific experimental design was used because the aim was to adjust the hardware components and configuration.

### 2.3. Hammer Milling

The extrudates were cooled and then conveyed through bucket elevators and collected in large plastic bags. The extrudates from these bags were then fed into the inlet of a Schutte Buffalo Hammer mill (Buffalo, NY, USA) fitted with a 3/64 in (1190 µm) screen. The powdered extrudates were filled directly into 50 lb (≈23 kg) 3-walled paper bags and sealed till further use.

### 2.4. Particle Size Analysis

The particle size of the extruded and milled binary blends was determined using a laser diffraction particle size analyzer (LSTM 13320, Beckman-Coulter, Inc., Miami, FL, USA). The samples were tested in duplicate. The equipment used the patented Polarization Intensity Differential Scattering technique to register the particle sizes. This technique is based on the concept that horizontal and vertical polarized light will have a small contrast for small particles, and this difference is used for the sizing of particles. Thus, when a sample passes through the laser beam, the difference in diffraction helps register the particle size.

### 2.5. Specific Mechanical Energy (SME)

SME measures the total mechanical energy input during the extrusion process. SME was calculated as follows (Equation (1)):(1)$$SME=\frac{\frac{\left(\tau -\tau_{0}\right)}{100}\times P_{rated}\times \frac{N}{N_{rated}}}{\dot{m}}$$
where *τ* = operating torque (%), *τ*_0_ = no-load torque (%), *P_rated_* = rated power (37.3 kW), *N* = screw speed (rpm), *N_rated_* = rated screw speed (507 rpm), and *ṁ* = net mass flow rate of extrudate at die exit (kg/s).

### 2.6. Protein and Fat Analysis

Standard methods were used to determine the proximate content of raw ingredients. The moisture content was determined using the AACC 44-19 method (135 °C for 2 h), crude protein was determined based on nitrogen by combustion (6.25X; AOAC 920.176), and crude fat was determined using the petroleum ether extract method (AOCS Ba 3-38). Duplicate analyses were run for each sample, and the results were reported on and as-is basis.

### 2.7. Mixing Protocol for Addition of Macro- and Micro-Ingredients

It was recommended by Webb et al. [11] in the FAQR that the solid content in the porridges made from FBFs be increased. The increase suggested by them was 20% solids as compared the current 11.75% [27] to ensure an energy-dense porridge. However, the porridge became thicker when the solid content was increased to this level, and it had difficulty flowing. The Bostwick flow rate for this new level was lower than the recommendation of 9–21 cm/min. In order to satisfy the Bostwick flow rate criteria, sugar was added at 15% after removing an equal quantity of the binary blend. The plasticizing and viscosity-reducing properties of sugar helped increase the Bostwick flow rate to within the recommendation. It has been reported that addition of sugar increases the energy density of FBFs with a minimum increase in volume [28].

The mixing was completed in steps of decreased dilution of ingredients as the steps progressed to ensure the uniformity of mixing. All the dry ingredients—sugar, WPC80, minerals, and vitamins—were weighed separately to create a batch of 25 kg of fortified blend. The minerals and vitamins were mixed first in a small Hobart mixer (Model N-50, Hobart Corporation, Troy, OH, USA) for 1 min until no yellow spots of vitamin were visible. This vitamin-mineral blend was then transferred into the bowl of another Hobart mixer (Model A-200, Hobart Corporation, Troy, OH, USA), and 3.33 kg of milled and extruded binary blend was added to it and mixed for 3 min to obtain a uniform dilution. This premix was then transferred slowly to the bowl of a larger Hobart mixer (Model M802, Hobart Corporation, Troy, OH, USA). Ten kg from the remaining milled binary blend was added and mixed for 5 min at a speed setting of 1. After 5 min of mixing the remaining dry ingredients, milled binary blends, sugar, and WPC80 were added and mixed for another 5 min at the same speed setting of 1. Once the mixing was over, 14.42 kg of the dry blend was removed from the bowl of the mixer. To the remaining 8.33 kg of dry blend remaining in the mixer, 2.25 kg of oil (for non-full-fat soy formulations) and 1.37 kg of oil (for full-fat soy formulations) were added and mixed for 5 min, with 2 min at a speed setting of 1 and 3 min at a speed setting of 2. On completion of this step, the previously removed dry blend was added back to the mixer and mixed for another 5 min at a speed setting of 1. At the end of this step, the fortified blended food was ready, and its composition is shown in Table 1.

### 2.8. Bostwick Consistometer Test (BCT)

BCT test was carried out using a Bostwick Consistometer (CSC Scientific Company, Inc., Fairfax, VA, USA). The Bostwick consistometer measures the flow of any viscous material, which in this case is the gruel/porridge made from the milled FBF, including the milled extrudates and other added macro- and micro-ingredients. The Bostwick consistometer is made of stainless steel and has a trough with precise 0.5 cm markings engraved along the base. At one end of the trough is a spring-loaded gate called the reservoir, which is the point at which the sample is loaded after closing the gate (Figure 2).

The consistometer was placed on a flat surface. Gruels containing 20% solids were made by adding 40 g of FBF to 160 mL of boiling distilled water in a glass beaker. After stirring for 1 min and cooling the gruel, it was then poured into the reservoir of the consistometer. After 30 s of settling, the gate of the reservoir was opened. The distance of flow was recorded exactly 1 min after opening the gate, and if the gruel was between two graduations at the end of 1 min, then the higher graduation was recorded as the distance (USDA 2005). All measurements were done in duplicate.

### 2.9. Water Absorption Index (WAI) and Water Solubility Index (WSI)

A method based on Anderson et al. [29] was used to measure the WAI and WSI of the binary blends before and after the extrusion process. A 2.5 g sample was dispersed in 25 g distilled water, using a glass rod to break any lumps. After stirring for 30 min, the dispersions were moved into tared centrifuge tubes; the volume was made up to 32.5 g by adding additional distilled water, and the samples were centrifuged at 3000 rpm for 10 min. The supernatant was decanted for its solid content (WSI) after evaporation of water from the supernatant. The sediment was weighed for its solid content and was used to report the WAI. The indices were calculated as shown in Equations (2) and (3):(2)$$\mathrm{WAI}\;(\%)=\frac{Weight\;of\;sediment}{Weight\;of\;dry\;solids}\times 100$$(3)$$\mathrm{WSI}\;(\%)=\frac{Weight\;of\;dissolved\;solids\;in\;supernatant}{Weight\;of\;dry\;solids}\times 100$$

### 2.10. Thermal Analysis—Differential Scanning Calorimetry (DSC)

Calorimetric measurements were carried out for different raw and extruded binary blends to understand the physical transformation of starch and proteins known as starch gelatinization and protein denaturation, respectively, on a Q100 DSC (TA Instruments, New Castle, DE, USA). Approximately 8–10 mg of sample was weighed into large-volume stainless steel DSC pans (Part No. 03190029, Perkin Elmer Health Sciences Inc., Shelton, CT, USA). Distilled water was added to the sample in the pan to obtain a solid-to-water ratio of 1:2 [30,31]. The pans were hermetically sealed, and the samples were allowed to equilibrate overnight. The instrument was calibrated using indium as the reference material. An empty sealed pan was used as a reference for all experiments. The program steps used for the test were as follows: Equilibrate at 10 °C, heat the pans from 10 °C to 140 °C at the rate of 10 °C/min, mark the end of the cycle, cool down the sample from 140 °C to 10 °C at the rate of 25 °C/min, mark the end of the cycle with a nitrogen gas flow rate of 50 mL/min. The samples were rescanned again by heating from 10 °C to 140 °C at the rate of 10 °C as the final phase of the test, to determine if there was any amylose-lipid complex peak, which would remain in the thermogram for the second heating run.

DSC data for each gelatinization and denaturation endotherm were analyzed for transition temperatures, onset (To), peak (Tp), endpoint or conclusion (Tc), and the enthalpy (∆H), using TA Instrument’s Universal Analysis Software (version 5.4.0). All the reported data were means of two replicates. Starch gelatinization (%) was calculated by comparing the enthalpic transition difference in starches between raw and extruded binary blends [32]. Calculations were made using the equation below (Equation (4)):(4)$$Starch\;geltinization\;\left(\%\right)=\frac{\Delta Hraw-\Delta Hextruded}{\Delta Hraw}\times 100$$
where ΔHraw = enthalpy of raw binary blend, and ΔHextruded = enthalpy of extruded binary blend. Total cook (%) was calculated as a ratio of the total enthalpic transition difference, which includes the transition enthalpies for starch and protein fractions of the binary blends. It is represented as below (Equation (5)):(5)$$Total\;cook\left(\%\right)=\frac{\Delta HTraw-\Delta HTextruded}{\Delta HTraw}\times 100$$
where ΔHTraw = total enthalpy of transition of raw binary blend, and ΔHTextruded = total enthalpy of transition of extruded binary blend.

### 2.11. Rapid Visco Analyzer (RVA)

RVA provides an index of how cooked a sample is by re-cooking under relatively low shear in excess water and measuring the pasting viscosity throughout the test. The pasting properties of the binary blends were examined using RVA (RVA 4, Newport Scientific Pvt. Ltd., Warriewood, NSW, Australia). Approximately 3.5 g of the sample was weighed and adjusted to a 14% moisture basis and was transferred to 25 mL of distilled water already placed in a canister. A plastic paddle was inserted into the canister and was rotated manually to disperse any lumps. The can along with the sample and the paddle was then attached to the RVA, and the test was initiated. The paddle initially rotated at 960 rpm for 10 s, and then it dropped down to 160 rpm for the remaining test duration. The RVA was interfaced with a computer equipped with Thermocline software for Windows (version 3.15.2.298) for controlling the test and analyzing the results. Pasting properties were determined after running the samples on a standard AACC profile (AACC 76-21.01, 1999) with a run time of 13 min. Peak viscosity (PV), pasting temperature (PT), trough or holding paste viscosity (HPV), breakdown (PV–HPV), final viscosity (FV), and setback (FV–HPV) were recorded. All measurements were performed in duplicate.

### 2.12. Starch Digestibility

Raw, extruded, and milled samples of the binary blends were measured by a modified in vitro Englyst method [33]. Samples of the blends (0.6 g) and guar gum (50 mg) were placed in a centrifuge tube (45 mL). Freshly prepared pepsin solution (50 mg pepsin in 10 mL 0.01 M HCl) was added to the tube, and the mixture was incubated for 2 h at 37 °C. Sodium acetate solution (0.25 M) was added to the mixture to stop the digestion. A pancreatic/amyloglucosidase mixture (5 mL) and glass beads were added to the sample tube for starch digestion. The tube was incubated in a shaking water bath at 37 °C and 90 strokes/min. At 20 and 120 min, an aliquot of 0.25 mL sample was pipetted into 10 mL of 66% ethanol solution. The glucose released at each interval was determined using a glucose oxidase/peroxidase method and was converted to a percentage of starch, hydrolyzed by multiplying by 0.9. Starch digested at 20 min is defined as RDS (rapidly digestible starch), starch digested between 20 and 120 min is defined as SDS (slowly digestible starch), and starch not digested after 120 min incubation is defined as RS (resistant starch).

### 2.13. Protein Digestibility

A protein digestibility assay modified from Mertz et al. [34] was used to measure the digestibility of the binary blends before and after extrusion. Flour samples (200 mg/sample) were weighed and placed in 50 mL centrifuge tubes. Each sample was incubated with 35 mL pepsin solution (1.5 mg pepsin in 1 mL of 0.1 M phosphate buffer pH 2.0) at 37 °C. After two hours of incubation, 2 mL of 2 M sodium hydroxide was added to each tube to stop the digestion. All tubes were centrifuged at 3320 g for 15 min at 4 °C, and the supernatant was discarded. The residue was washed in 10 mL of 0.1 M phosphate buffer pH 2.0 and centrifuged, and the supernatant was discarded. The washing steps were repeated one more time, and samples were frozen (−80 °C for 30 min) and lyophilized. Freeze-dried samples were tested using nitrogen combustion (LECO system) to analyze the amount of undigested protein. Digested protein was calculated based on the protein content of the native sorghum flour and that of the undigested fraction (Equation (6)).(6)$$Digestibility\left(\%\right)=\frac{Total\;N\;\left(mg\right)-Undigested\;N\;(mg)}{Total\;N\;(mg)}\times 100$$

### 2.14. Phytic Acid

Phytic acid (InsP6) was measured using a Megazyme kit (Megazyme International, Wicklow, Ireland) for phytic acid and total phosphorous, in which phytic acid was measured as the phosphorous released by phytase and alkaline phosphatase. One gram of powdered sample was transferred into a 75 mL glass beaker containing 20 mL of 0.66 M hydrochloric acid. The beaker was covered with aluminum foil and stirred vigorously overnight at room temperature. One mL of extract was transferred to a 1.5 mL microcentrifuge tube and centrifuged at 13,000× *g* for 10 min. A total of 0.5 mL of the resulting extract was immediately transferred to a fresh 1.5 mL microcentrifuge tube and neutralized by the addition of 0.5 mL of 0.75 M of sodium hydroxide solution. A neutralized sample extract solution was used in the enzymatic dephosphorylation reaction procedure. Total and free phosphorous contents were measured for each sample. ∆A phosphorous was calculated for each sample by subtracting the absorbance of the ‘free phosphorous’ sample from that of the absorbance of the ‘total phosphorous’ sample. The phosphorous concentration is expressed as g/100 g of sample using the following calculation (Equation (7)):(7)$$Phosphorus\;(g/100g)=\frac{\Delta A\;phosphorous\times 10,000\times 1.0\left(original\;sample\;wt.\right)\times 1.0\;}{mean\;M\times 20\times 55.6}$$
where 10,000 = conversion value from µg/g to g/100 g; wt. = weight; 1.0 = sample volume used in colorimetric step; mean M = mean value of phosphorous standard; 20 = original sample extract volume; 55.6 = dilution factor.

The mean value of phosphorous standard was obtained using a standard curve over a dynamic range of 0–0.75 µg phosphorous. Phytic acid concentration was calculated as follows (Equation (8)):(8)$$Phytic\;acid\;(g/100g)=\frac{Phosphorous\;(g/100g)}{0.282}$$

The calculation of phytic acid content was based on the assumption that the amount of phosphorous measured is exclusively released from phytic acid and that this comprises 28.2% phytic acid.

### 2.15. Tannins

The Vanillin-HCl method of Price et al. [35] was used to measure the tannins. The test samples were ground, and 0.3 g was taken for assessing the tannins. The ground samples were transferred into 15 mL tubes. The extraction of tannins was performed by adding 8 mL of 1% HCl in methanol at 1 min intervals to the sample tubes. The sample in the tube with 1% HCl in methanol was vortexed for 10 s and then placed in a water bath maintained at 30 °C for exactly 20 min. The sample tubes were removed from the water bath after 20 min and vortexed again. The extracts were then centrifuged at 2000 rpm for 4 min. Two 1 mL aliquots were taken from each tube and transferred into two separate 15 mL tubes. One tube was marked for sample determination, and the other tube was marked for blank determination. The sample tubes after brief vortexing were placed in a water bath maintained at 30 °C after adding 5 mL of vanillin reagent at 1 min intervals. Then, 1% vanillin in methanol was added to 8% HCl in methanol in a 1:1 ratio to prepare the vanillin reagent. The blank tubes were added with 5 mL of 4% HCl in methanol and, after brief vortexing, were placed in a water bath at 30 °C. After 20 min, the absorbance of samples and blanks was read using a spectrophotometer at 500 nm. The spectrophotometer was zeroed using a methanol blank before measuring. The difference between sample and blank was used for final determination of the phenol content and expressed as catechin equivalent (CE)/mg of sample. A standard curve using 1000 ppm of catechin standard at 0, 0.2, 0.4, 0.6, 0.8, and 1.0 mL added with 1% HCl in methanol to make up the volume to 1 mL and 5 mL vanillin reagent was plotted. The CE equivalent was calculated as below (Equation (9)):(9)$$CE\;(mg/mg\;of\;sample)=\frac{\left(y/m\right)}{ample\;concentration\;(mg/mL)}$$
where *y* and *m* (slope) can be calculated from the regression equation of the standard curve.

### 2.16. Trypsin Inhibitor

Trypsin inhibitor activity was determined according to the method described by Smith et al. [36], using BAPNA (benzoyl-DL-arginine-*p*-nitroanilide) as a substrate (0.92 mM in 0.05 M Tris buffer/0.02 M CaCl2, pH 8.2). The sample was ground into fine sizes, and 1 g was used for extraction of trypsin in 50 mL of 0.01 N NaOH for 3 h. The pH was maintained within the range of 8.5 to 9.0. One mL of the extract and 2 mL of trypsin solution 0.002% (Type I, of bovine pancreas, SIGMA, Setagaya City, Tokyo, Japan) in 0.001 M HCl were mixed with 1 mL of water. The reaction started after the addition of 5 mL of substrate at 37 °C. After 10 min, the reaction was stopped by the addition of 1 mL of 30% acetic acid. The reaction mixture was filtered through filter paper (Whatman No. 3), and the absorbance due to the release of *p*-nitroaniline was read at 410 nm. The activity was interpreted as the increment of 0.01 units of absorbance at 410 nm for 10 mL of reaction mixture. Trypsin inhibitor activity is expressed in terms of mg trypsin inhibited per g of dry sample.

### 2.17. Statistical Analysis

All the results were analyzed using one-way analysis of variance (ANOVA) with a general linear model procedure (SAS version 9.1, SAS Institute, Cary, NC, USA). When significant effects (*p* < 0.05) were indicated by ANOVA, Tukey pairwise comparisons were performed to identify which treatments differed significantly (*p* < 0.05).

## 3. Results and Discussion

### 3.1. Particle Size of Milled Extrudate

The mean particle size of the extruded binary blends of SSBs ranged from 196.93 µm to 424.28 µm (Table 2), with the lowest being SS″B-V1 (commercial) and SS″B-V2. Amongst the commercial blends, the whole grain formulation had a significantly (*p* < 0.05) bigger average particle size than the decorticated samples. In the pilot blends, whole formulations had significantly lower (*p* < 0.05) particle sizes as compared to decorticated blends. No significant correlations were observed between different extrudates′ physico-chemical properties with respect to particle size. However, the effect of overall fat content in the blend, which varied due to the presence of different types of soy flour in the binary blend, seemed to affect the particle size when the commercial and pilot blends were analyzed separately. Higher fat content produced extrudates with higher density due to low expansion, and when subjected to hammer milling, these extrudates needed more force than other expanded blends to disintegrate; hence, they produced lower particle sizes. Despite the differences in particle sizes of the blends, they all met the particle size specifications of the USDA commodity requirements for CSB Plus [37], which is currently being distributed in food aid programs.

### 3.2. Water Absorption Index (WAI) and Water Solubility Index (WSI)

It was observed from Table 2 that the WAI for SSBs ranged from 2.85 g/g to 5.91 g/g for WSS″B-V1 and SS″B-V2 (both pilot milled), respectively. From the Table 2, it is also seen that WSI ranged from 6.23% in WSSB-V1 to 21.45% in WSS″B-V1 (both pilot milled). The WAI and WSI data did not exhibit a definitive trend based on the type of formulation or the type of milling from which these formulations were obtained. The impact of formulation affected the WAI. From Figure 3, it can be observed that WAI was positively correlated to starch content (r = 0.78). Higher starch content leads to higher water absorption due to higher gelatinization, and an increase in SME leads to an increase in WAI, as shown in Figure 4a (r = 0.66).

WAI increased when SME was increased due to an increase in the amount of swollen starch in the undamaged starch granules [38]. This may seem counterintuitive, because the general phenomenon that is observed is that higher SME would cause higher shear and cause damage to the starch polymer; thus, the availability of undamaged starch is reduced. This causes the lowering of WAI. However, in this study it was observed that the formulations with higher soy fat content had higher SME, and the presence of higher fat may have protected the starch breakdown regularly associated with higher SME and thus higher WAI. The fiber content in the blend was inversely correlated to WAI (r =−0.76), as shown in Figure 4b. Chevanan et al. [39] reported that increasing DDGS content, which is rich in fiber, from 20 to 60% (which in turn reduced the starch content of the sample), reduces the WAI of extrudates by 25.7%. There was an inverse correlation between WAI and WSI; higher WAI is indicative of less starch degradation and thus it lowers the WSI. Besides the starch gelatinization, which results in the release of amylose and amylopectin, starch dextrinization and other reactions can also occur, leading to the formation of low-molecular-weight compounds and influencing the WSI [40].

### 3.3. Degree of Starch Gelatinization

The starch gelatinization levels of SSB formulations are summarized in Table 2. The extrusion processing aided in starch gelatinization, which ranged from 90.70 to 96.27%. There was no statistically significant difference (*p* > 0.05) in gelatinization values across different blends, except between SS″B-V2 and WSS″B-V1. However, it was noted that blends with higher oil content (whole blends) had lower values of gelatinization. Similarly, the total cook of the blends after taking into consideration the total enthalpy (starch gelatinization and protein denaturation) ranged from 64.46 to 78.50%, as shown in Table 2. Although all the blends had starch gelatinization above 90%, the differences in the blends could be attributed to the levels of starch in the blends. The starch content in whole blends ranged from 47.36 to 54.71% and that in decorticated blends ranged from 53.71 to 57.28%. Thus, higher starch content could have contributed towards higher transition enthalpy during the starch gelatinization process. Also, lower inherent fat content in decorticated blends formed lower amylose-lipid complexes during the thermal transition process, thereby increasing the gelatinization when compared to whole blends. Similarly, whole blends had higher protein content (13.37–19.00%) and higher fat content (1.30–3.32%) as compared to decorticated blends (12.87–18.70% protein and 2.85–6.62% inherent fat (data not presented)). This accounted for lower energy transfer to whole blends during processing because oil provides a lubricating effect [41] and this would have caused the protein, which requires higher energy, to be less denatured than starch when cooked as compared to decorticated blends. Harper et al. [42] reported that the heat of reaction for protein was 90–100 kJ/kg and for cereal starches was 10–19 kJ/kg [30].

### 3.4. Pasting Properties

The pasting properties of ground-extruded binary blends of SSB are shown in Table 3. The results in the table show that blends containing whole grain sorghum flour had higher peak time compared to formulations that were constituted with decorticated sorghum flour, though these were not significantly different. Peak time indicates the time span required to reach peak viscosity, where the starch granules swell until the starch granules rupture [43]. The high fiber content in whole grain blends competes with starch in water absorption and therefore delays the swelling of starch, leading to higher peak time. In a study by Yildiz et al. [44], it was found that the blending of wheat starch with oat and pea fibers led to longer peak time. Also, higher SME lowered the peak time (Figure 4c) due to its effect in breaking down starch molecules (r = −0.46).

The peak viscosity was higher in blends with decorticated sorghum flour, though the difference was not significant. The whole sorghum-based blends had higher protein and fat, which affected the peak viscosity. Egouletey and Aworh [45] reported that protein and fat interaction in the blend of African yam, beans, and cassava starch lowers the peak viscosity. Sandhu and Singh [46] also attributed lower peak viscosity to difference in the protein contents of blends. The breakdown viscosity range was 17.50–272.50 cP (0.17–0.27 Pa.s). Trough and breakdown pasting properties indicate the ability of a food material to remain undisrupted on being subjected to long periods of constant high temperature and the ability to withstand breakdown during cooking [47]. It is regarded as the measure of the degree of disintegration of the granules or “paste stability” [48]. It was observed that blends with whole sorghum had lower breakdown viscosities than decorticated blends. Even though the troughs did not generally show any significant difference (*p* > 0.05) between blends with whole and decorticated sorghum and with different levels of fat, it was observed that fiber and fat had an impact on the breakdown viscosities. SS″B-V1 (aspirated) showed the highest breakdown viscosity, which meant it was the least stable during the holding phase; whole blends were more stable during the holding phase as compared to decorticated blends owing to their low peak viscosities and lower starch content. The final viscosity ranged from 153.50 to 260.00 cP (0.15–0.26 Pa.s). Shimels et al. [49] reported that final viscosity was used to indicate the ability of starch to form a paste or gel after cooling. Higher final viscosities were observed in blends containing whole flour and/or high-fat content, though the difference was not significant (*p* > 0.05).

The low peak viscosity and breakdown viscosity and the high final viscosity of whole grain blends could be linked to the degree of amylose leaching, amylose–lipid complex formation, friction between swollen granules, granule swelling, and competition for free water between leached amylose and the remaining ungelatinized granules [50]. This result appeared to suggest that the starch granules became more resistant to thermal treatment and mechanical shearing [51]. A decrease in breakdown viscosity might be due to less swelling of the starch granules in the presence of fiber [52] and fat. The higher final viscosities in whole grain blends may have been the combined effect of retrogradation of amylose molecules as well as higher water absorption by the inherent fiber content in the blend. The peak viscosities for binary blends containing whole sorghum flour were significantly lower (*p* < 0.05) than those of the blends with decorticated sorghum flour. The extruded binary blends containing decorticated sorghum flour exhibited lower setback viscosity values as compared to whole sorghum blends.

### 3.5. Bostwick Consistency

The Bostwick flow rates of the cooked slurry at 20% solid concentration of the extruded binary blend, which had sugar, WPC 80, oil, vitamins, and minerals, is presented in Table 4. It is observed that whole formulations and blends with higher fat content had lower Bostwick values.

Further, it is observed from Figure 5 that there was a positive correlation between ER and Bostwick Flow rate (r = 0.81). ER is the overall effect on the extrudate, resulting from the formulation and the process SME. Higher ER results from higher SME and lower fat content in the formula. Therefore, higher ER is indicative of higher starch transformation due to its breakdown during the process of extrusion. The starch breakdown produces lower molecular fractions, which do not absorb as much water as an intact starch granule [53] and therefore flow more freely during the Bostwick flow test. The final viscosity of blends during RVA did not correlate with the Bostwick flow rate, possibly because the Bostwick flow rate was conducted at 30 °C, whereas final viscosity in the RVA was recorded at 50 °C. This difference in temperature might have further changed the viscosity profile of the blends [54]. Despite the differences in flow rate between all the blends, they were within the range of 9–21 cm/min as stipulated by the USDA commodity requirement for corn-soy blends (CSB 13) [27], except for WSS″B-V2, which had the lowest SME, at 51.20 kJ/kg, and an ER of at least 2.43. Analysis of all these correlations highlights the fact that Bostwick flow rate is dependent on the characteristics of starch after it has been cooked as porridge/slurry.

### 3.6. Starch and Protein Digestibility

The in vitro starch and protein digestibility studies were conducted on a select formulation of SSB. The formulation was selected based on the energy density of each binary blend and the superior stability, i.e., relatively fewer changes in operational torque during extrusion processing. This eliminated the whole blends and low- and high-fat blends from this part of the study. SS′B-V1 was chosen from the binary blend made from commercially sourced sorghum flour.

It is observed from Table 5 that the RDS saw a significant increase (*p* < 0.05), from 10.29% to 25.69%, in binary blends after extrusion as compared to the raw blend. The RDS increased by 149.65% after extrusion because the maximum starch breakdown/cooking occurs inside the extruder barrel, which was discussed earlier in the DSC section. There was a 21.53% decrease in SDS, which was significant (*p* < 0.05), in the extruded samples as compared to raw blend. Further, it was observed that there was a decrease in RDS after cooking the extruded samples, whereas the SDS increased. It has been reported that wet cooking of sorghum reduces its digestibility [55,56]. The starch, in its native state, is also referred to as slowly digestible starch because it can be digested in the small intestine, albeit slowly [33].

Starch needs to be gelatinized for efficient hydrolysis and subsequent bioavailability, since gelatinized starch is more susceptible to enzymatic attack [57]. In vitro starch digestibility significantly improved after the raw blend was extruded. As the processing/cooking progressed, there was a decline in SDS. It then increased for the SS′B binary blend, where it could be either seen as an increase in RDS or RS, depending on the nature of the starch under study. In SSBs, higher SDS after cooking resulted in higher RS. This could have been the result of a higher degree of retrogradation when wet-cooked. On comparison to control samples CSB13 and CSB+, it was found that raw controls had higher RDS and SDS than SSBs. The lower RS in CSB13 and CSB+ after cooking it as porridge could have been due to the lower amount of retrogradation in the control samples as compared to SSBs.

The protein digestibility for SSB binary blends is shown in Table 6. It was observed from the table that there was no significant difference in protein digestibility between the raw and extruded binary blends of SSB. The raw blend had a high digestibility (87.21%) compared to the extruded, with 86.69%. There was no significant difference (*p* > 0.05) between the protein digestibility of raw SSB, CSB13, and CSB+. However, after cooking, the digestibility of the SSB decreased significantly (*p* < 0.05) to 82.17%. The digestibility of CSB13 and CSB+ increased after cooking, to 88.35% and 89.22%, though the increase was not significant. It was found that the decrease in protein digestibility of the SSB after cooking was lower than that of CSB13 and CSB+, but the difference was not significant (*p* > 0.05). Previous studies have shown a significant increase in protein digestibility after extrusion [58,59,60,61]. The basic premise for the above studies and other similar findings has been that extrusion causes protein denaturation, which increases its susceptibility to enzymatic hydrolysis and therefore improves digestibility [62]. The results of the SSB are not in confirmation with the above findings and could have been caused by cross-linking reactions for protein–protein, starch–protein, and protein–lipid. During extrusion, however, some interactions can reduce digestibility due to non-enzymatic browning reactions and the formation of cross-linking reactions [40,63,64,65,66,67,68]. Onwulata et al. [69] reported that the extrusion process does not affect the overall protein percentage. Furthermore, these researchers reported that although the amount of denatured protein increased due to extrusion temperatures, this denaturation had a minimal overall effect on protein digestibility.

### 3.7. Anti-Nutritional Factors

The levels of phytic acid in the raw and extruded SSB blends are presented in Table 7. It was found that there was a significant difference (*p* < 0.05) between phytic acid content in raw blends as well among extruded blends. The mean phytic acid content in the raw and extruded blends of SSB was found to be 752.80 mg/100 g and 556.6 mg/100 g, respectively. Thus, there was a significant (*p* < 0.05) reduction of 26.06% in phytic acid levels after extrusion. A significant reduction in phytic acid and tannins after the extrusion of acha/soybean flours was reported by Anuonye et al. [70]. Batista et al. [71] reported a similar level of reduction in phytic acid content (17–26%) after the extrusion of common beans (*Phaseolus vulgaris* L.). High-performance liquid chromatography (HPLC) analysis of low-fiber diets of bran-gluten-starch mixtures after extrusion showed that some molecules of inositol hexaphosphate were hydrolyzed to penta-, tetra-, and triphosphates due to thermal degradation [72]. Barroga et al. [73] and Kataria et al. [74] determined thermal degradation changed these molecules along with their chemical reactivity and that the formation of insoluble complexes could explain the reduction in the anti-nutrient content due to thermal processing. Thus, the heat and shear inside the extruder can be used to improve the nutritional value of cereal blends by reducing phytic acid levels.

Tannin, a phenolic derivative of flavone, occurs as glycosides in the natural state, forming complexes with available protein [75,76] and inhibiting enzyme [77] activity, which affects digestibility. No tannins were found in raw binary blends (Table 7). Individual analysis of each raw component of the binary blend also did not show any presence of tannins. Most of the U.S.-grown sorghums do not contain tannins [15,78]. Less than 2% of the sorghum grown in the U.S. has high tannins [79]. Egounlety and Aworh [80] reported the absence of tannins in dehulled soybeans.

Trypsin inhibitors are enzymes that impede the activity of trypsin, a digestive enzyme that is used by the body to breakdown proteins, therefore affecting the ability to digest and absorb dietary proteins [81]. Trypsin inhibitor levels were reduced significantly (*p* < 0.05) after extrusion in SSB when compared to raw samples (Table 7). A reduction of 19.50% in trypsin inhibitor levels was observed after extrusion. The trypsin inhibitor is heat labile and degrades significantly with extrusion processing [82]. Other studies have also found that trypsin inhibitors are thermolabile and their inhibitory activity can be reduced by an appropriate thermal treatment [58,83,84]. Balandran-Quintana et al. [59] observed that extrusion cooking was one of the best processing methods for improving the protein quality of legumes.

### 3.8. Energy Density of FBFs

FBFs are used in infant feeding programs as complementary foods, provided at a suitable age to meet growth requirements. The FAQR [11] recommended energy-dense foods with good protein content and an appropriate inclusion of essential micronutrients as necessary (albeit not always sufficient) to achieve defined nutrition goals among vulnerable populations. Therefore, they advised that the current nutritional and energy value of the FBFs, CSB or WSB, be enhanced to provide about 387 kilocalories as energy, 18 g protein, and 9 g fat in 100 g of FBF served. They further suggested the use of other cereal blends than the regular CSB and WSB as well as the use of WPC to increase the protein content in addition to other changes. In addition to these, the availability of staple foods must be ensured so that nutritionally enhanced foods are in addition to rather than replacing these as sources of energy and nutrition in the local food supply.

In the current study, it is observed (Table 4) that all SSB formulations had an energy density in the range of 394.36–413.63 kcal/100 g from a 20% concentration based on the FAQR 2011 [11] recommendations, which is comparatively higher than previous FBFs used in feeding programs. The CSB Plus being programmed by the USDA currently stipulates a minimum energy density requirement of 380 kcal/100 g [37], whereas the specification for Super Cereal Plus is a minimum of 410 kcal/100 g [85].

## 4. Conclusions

Based on our findings, this study demonstrates that raw material composition and processing conditions significantly influenced the physicochemical and functional properties of fortified blended foods (FBFs). The inherent fat content, starch, and fiber levels in the blends play critical roles in determining key attributes,, such as particle size, water absorption index (WAI), water solubility index (WSI), and Bostwick flow rate. Extrusion processing not only enhanced starch gelatinization and protein digestibility but also effectively reduced anti-nutritional factors, such as phytic acid, tannins, and trypsin inhibitors. Furthermore, the energy density of the FBFs exceeded the USDA recommended standards, ensuring their nutritional adequacy. Notably, the use of whole sorghum flour in blends resulted in unique rheological properties, such as lower Bostwick flow rates and higher final viscosities, which may have implications for their practical applications. These results highlight the potential of optimizing raw material composition and extrusion processing parameters to develop FBFs with improved nutritional and functional properties, catering to diverse dietary and health needs. Future studies will explore the outcomes from other binary blends of grains and legumes and elucidate the similarities and differences between the FBFs.

## Figures and Tables

**Figure 1 foods-14-00779-f001:**
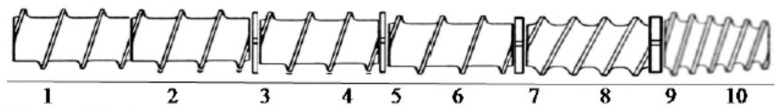
Schematic of the extruder showing the screw profile. Screw elements: 1–2 = single flight screws, 3 = small steam lock, 4 = single flight screw, 5 = small steam lock, 6 = single flight screw, 7 = medium steam lock, 8 = double flight screw, 9 = large steam lock, 10 = triple flight uncut cone screw.

**Figure 2 foods-14-00779-f002:**
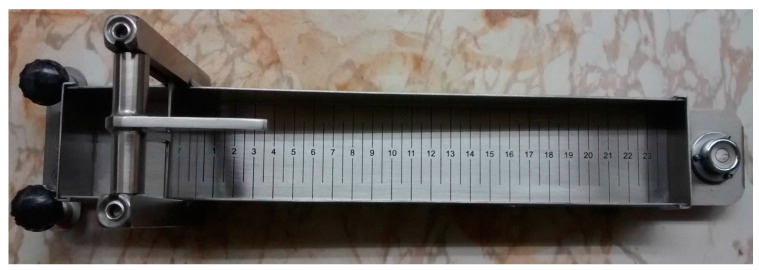
Image of Bostwick consistometer.

**Figure 3 foods-14-00779-f003:**
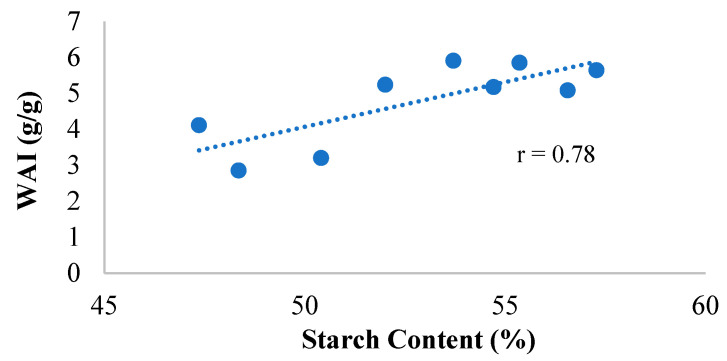
Correlation between starch content and WAI for binary blends of SSB. WAI = water absorption index; SSB = sorghum-soy blend.

**Figure 4 foods-14-00779-f004:**
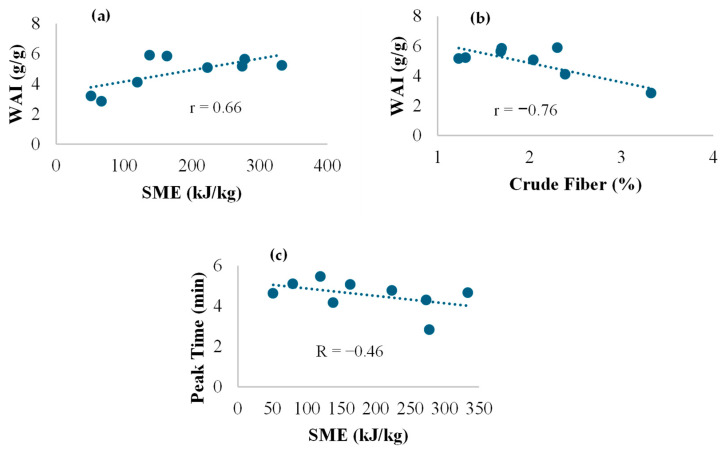
Correlations between (**a**) SME and WAI, (**b**) crude fiber and WAI, and (**c**) SME and peak time for binary blends of SSB. SME = specific mechanical energy; WAI = water absorption index; SSB = sorghum-soy blend.

**Figure 5 foods-14-00779-f005:**
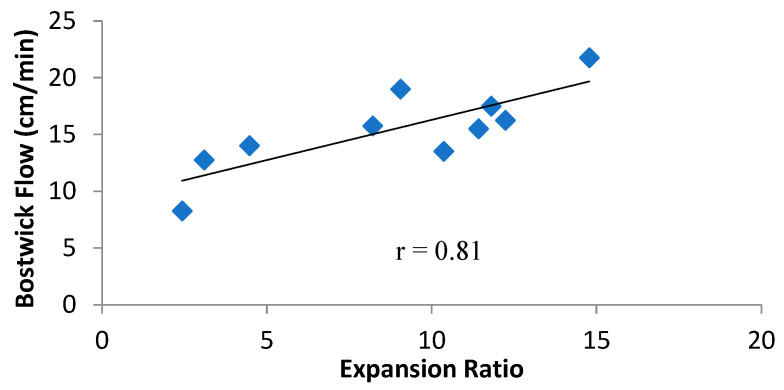
Correlation between expansion ratio (ER) and Bostwick flow rate of fortified blended SSB. ER = expansion ratio; SSB = sorghum-soy blend.

**Table 1 foods-14-00779-t001:** Composition of SSB-based FBF.

Ingredients	Amount (%)
Milled extrudates (SSB)	63.40
Sugar	15.00
WPC80	9.50, 13.00 (for high-fat blends)
Oil	9.00, 5.50 (for high-fat blends)
Mineral Premix	3.00
Vitamin Premix	0.10

SSB = sorghum-soy blend; FBF = fortified blended food; WPC80 = whey protein concentrate with 80% protein.

**Table 2 foods-14-00779-t002:** Mean values of SME, particle size, WAI, WSI, starch gelatinization, and total cook of SSB formulations.

Formulation	SME (kJ/kg)	Average Particle Size (µm)	WAI (g/g)	WSI (%)	Starch Gelatinization (%)	Total Cook (%)
SS′B-V1 com	273 ± 28 ^ab^	208.52 ± 0.83 ^f^	5.17 ± 0.31 ^ab^	7.79 ± 0.99 ^b^	93.62 ± 3.90 ^ab^	77.51 ± 3.76 ^a^
SS″B-V1 com	223 ± 12 ^abc^	196.93 ± 4.98 ^g^	5.08 ± 0.26 ^b^	8.72 ± 0.39 ^b^	92.10 ± 2.21 ^ab^	71.57 ± 4.91 ^ab^
WSS′B-V1 com	333 ± 52 ^a^	224.02 ± 0.25 ^e^	5.23 ± 0.24 ^ab^	7.23 ± 0.48 ^b^	93.06 ± 2.44 ^ab^	68.50 ± 1.37 ^ab^
WSS″B-V1	67 ± 25 ^d^	230.27 ± 1.14 ^e^	2.85 ± 0.18 ^d^	21.45 ± 2.06 ^a^	90.69 ± 0.43 ^b^	64.46 ± 1.87 ^b^
WSSB-V1	119 ± 55 ^cd^	255.43 ± 3.85 ^d^	4.11 ± 0.41 ^c^	6.23 ± 0.62 ^b^	94.47 ± 1.42 ^ab^	69.34 ± 4.16 ^ab^
SS″B-V1	163 ± 42 ^bcd^	388.15 ± 3.22 ^b^	5.85 ± 0.23 ^ab^	6.82 ± 0.88 ^b^	94.06 ± 0.17 ^ab^	78.44 ± 6.13 ^a^
WSS″B-V2	51 ± 21 ^d^	255.08 ± 2.63 ^d^	3.20 ± 0.13 ^d^	9.13 ± 0.41 ^b^	92.84 ± 1.44 ^ab^	66.82 ± 4.09 ^b^
SS″B-V2	138 ± 72 ^cd^	424.28 ± 2.18 ^a^	5.91 ± 0.18 ^a^	7.30 ± 0.93 ^b^	96.26 ± 0.31 ^a^	78.50 ± 0.88 ^a^
SS″B-V1 (aspi)	278 ± 65 ^ab^	375.59 ± 1.85 ^c^	5.64 ± 0.41 ^ab^	18.51 ± 2.84 ^a^	95.08 ± 0.56 ^ab^	78.22 ± 0.05 ^a^

Note: com = blends that had raw materials procured commercially; for other blends, the raw flours were produced by milling at the pilot mill. Aspi = full-fat soy. Values in the same column not sharing the same superscript are significantly different at *p* < 0.05. SME = specific mechanical energy; WAI = water absorption index; WSI = water solubility index; First S = decorticated sorghum flour; 2nd S = low-fat soy flour; *S′* = medium-fat soy flour; *S″* = high-fat soy flour; W = whole; V1 and V2 = white varieties of sorghum.

**Table 3 foods-14-00779-t003:** Mean pasting properties of binary blends of SSB.

Formulation	PV (cP)	Peak Time (min)	PT (°C)	Trough (cP)	Breakdown (cP)	FV (cP)	Setback (cP)
SS′B-V1 com	232.00 ± 1.41 ^cd^	4.31 ± 0.09 ^ef^	72.55 ± 1.20 ^a^	108.50 ± 10.61 ^c^	123.50 ± 9.19 ^b^	156.00 ± 33.94 ^e^	47.50 ± 23.33 ^bc^
SS″B-V1 com	257.50 ± 7.78 ^bc^	4.78 ± 0.00 ^bcd^	73.40 ± 0.07 ^a^	151.50 ± 2.12 ^b^	106.00 ± 5.66 ^bc^	170.00 ± 7.07 ^de^	18.50 ± 9.19 ^c^
WSS′B-V1 com	191.50 ± 10.61 ^d^	4.68 ± 0.24 ^cde^	72.20 ± 2.90 ^a^	135.00 ± 28.28 ^bc^	56.50 ± 38.89 ^cde^	177.50 ± 30.41 ^cde^	42.50 ± 2.12 ^bc^
WSS″B-V1	144.00 ± 18.38 ^d^	5.11 ± 0.19 ^ab^	68.37 ± 4.77 ^a^	127.50 ± 10.61 ^bc^	16.50 ± 7.78 ^e^	260.50 ± 35.36 ^a^	127.50 ± 31.82 ^a^
WSSB-V1	277.50 ± 17.68 ^b^	5.48 ± 0.04 ^a^	70.52 ± 4.00 ^a^	211.50 ± 20.51 ^a^	66.00 ± 2.83 ^bcde^	239.50 ± 19.09 ^abc^	28.00 ± 1.41 ^c^
SS″B-V1	193.50 ± 12.02 ^d^	5.08 ± 0.04 ^abc^	56.80 ± 9.62 ^bc^	150.50 ± 14.85 ^bc^	43.00 ± 26.87 ^de^	243.50 ± 16.26 ^ab^	93.00 ± 31.11 ^ab^
SS″B-V2	233.00 ± 19.80 ^cd^	4.18 ± 0.10 ^f^	50.00 ± 0.00 ^c^	139.00 ± 14.14 ^bc^	94.00 ± 5.66 ^bcd^	194.50 ± 17.68 ^bcde^	75.00 ± 24.04 ^abc^
WSS″B-V2	147.50 ± 6.36 ^d^	4.64 ± 0.16 ^de^	66.85 ± 2.05 ^ab^	130.00 ± 2.83 ^bc^	17.50 ± 3.54 ^e^	222.50 ± 17.68 ^abc^	92.50 ± 20.51 ^ab^
SS″B-V1 (aspi)	397.50 ± 24.75 ^a^	2.84 ± 0.19 ^g^	50.00 ± 0.00 ^c^	125.50 ± 11.31 ^bc^	272.50 ± 36.06 ^a^	153.50 ± 7.78 ^e^	28.50 ± 3.54 ^c^

Note: com = blends that had raw materials procured commercially; for other blends, the raw flours were made by milling at the pilot mill. PV = Peak Viscosity. PT = Pasting Temperature. FV = Final Viscosity. Values in the same column not sharing the same superscript are significantly different at *p* < 0.05. First S = decorticated sorghum flour; 2nd S = low-fat soy flour; S′ = medium-fat soy flour; S″ = high-fat soy flour; W = whole; V1 and V2 = white varieties of sorghum; aspi = aspirated full-fat soy flour.

**Table 4 foods-14-00779-t004:** Bostwick flow rate, crude protein, crude fat, and energy density of binary blends and fortified blends of SSB.

Formulation	Bostwick Flow Rate (cm/min)	Binary Blend Protein (%)	FBF Protein (%)	Binary Blend Fat (%)	FBF Fat (%)	Binary Blend Energy Density (kcal/100 g)	FBF Energy Density (kcal/100 g)
SS′B-V1 com	17.50 ± 0.71 ^c^	18.70	19.45	3.67	12.09	372.27	413.63
SS″B-V1 com	13.50 ± 0.71 ^e^	12.93	18.60	6.17	10.45	382.75	402.93
WSS′B-V1 com	16.30 ± 0.35 ^cd^	18.53	19.35	3.94	12.25	371.84	413.36
WSS″B-V1	12.80 ± 0.35 ^e^	14.60	19.66	6.62	10.74	374.90	397.94
WSSB-V1	14.00 ± 0.00 ^e^	19.00	19.65	2.85	11.57	357.20	402.97
SS″B-V1	15.80 ± 0.35 ^d^	13.12	18.72	5.59	10.09	373.99	397.36
WSS″B-V2	8.30 ± 0.35 ^f^	13.37	18.88	6.75	10.82	376.04	398.66
SS″B-V2	19.00 ± 0.00 ^b^	13.83	19.17	5.21	9.84	369.27	394.36
SS″B-V1 (aspi)	21.80 ± 0.35 ^a^	12.87	18.56	4.65	9.49	370.32	395.03

FBF = fortified blended food; com = blends that had raw materials procured commercially; for other blends, the raw flours were made by milling at the pilot mill. First S = decorticated sorghum flour; second S = low-fat soy flour; S′ = medium-fat soy flour; S″ = high-fat soy flour; W = whole; V1 and V2 = white varieties of sorghum; Aspi = aspirated full-fat soy flour. Values in the same column not sharing the same superscript are significantly different at *p* < 0.05. FAQR [11] guidelines for FBFs: Protein ≥ 18 g, Fat ≥ 9 g, Energy ≥ 387 Kcal. The proximate content numbers are theoretically values; they were calculated from the proximate content of the ingredients.

**Table 5 foods-14-00779-t005:** Starch digestibility of binary blends of SSB.

Binary Blend	RDS (%)	SDS (%)	RS (%)
SS′B–RM	10.29 ± 0.23 ^f^	41.48 ± 1.11 ^d^	48.23 ± 0.92 ^a^
SS′B–ME	25.69 ± 0.39 ^b^	32.55 ± 0.80 ^f^	41.76 ± 1.05 ^c^
SS′B-Cooked	18.48 ± 0.17 ^e^	36.87 ± 1.03 ^e^	44.65 ± 0.95 ^b^
CSB13 RM	18.72 ± 0.18 ^e^	54.66 ± 0.07 ^a^	26.62 ± 0.14 ^f^
CSB13 Cooked	21.42 ± 0.31 ^d^	49.79 ± 0.64 ^b^	28.79 ± 0.35 ^e^
CSB+ RM	23.28 ± 0.50 ^c^	47.17 ± 0.98 ^c^	29.55 ± 0.64 ^e^
CSB+ Cooked	27.37 ± 0.26 ^a^	41.46 ± 0.39 ^d^	31.17 ± 0.47 ^d^

SS′B = sorghum-soy blend (commercial milled); CSB = corn-soy blend; RM = raw material; ME = milled extrudate; RDS = rapidly digestible starch; SDS = slowly digestible starch; RS = resistant starch. Values in the same column not sharing the same superscript are significantly different at *p* < 0.05.

**Table 6 foods-14-00779-t006:** Protein digestibility of SSB binary blend, CSB13, and CSB Plus.

Formulation	RM-Digestibility (%)	ME-Digestibility (%)	Cooked Digestibility (%)
SS′B-V1com	87.21 ± 6.60 ^a^	86.69 ± 1.74 ^a^	82.17 ± 3.61 ^a^
CSB13	80.72 ± 1.00 ^a^	N/A	88.35 ± 4.06 ^a^
CSB Plus	85.80 ± 0.40 ^a^	N/A	89.22 ± 3.96 ^a^

SS′B = sorghum-soy blend (commercially milled); CSB13 = corn-soy blend13; CSB Plus = corn-soy blend plus; RM = raw material; ME = milled extrudate; com = blend that had raw materials procured commercially; V1 = white sorghum variety. Values in the same row not sharing the same superscript are significantly different at *p* < 0.05.

**Table 7 foods-14-00779-t007:** Comparison of changes in phytic acid, tannins, and trypsin inhibitor before and after extrusion of SSB with those of raw CSB Plus.

Formulation	Phytic Acid (mg/100 g)		Tannins (mg/100 mg CE)		Trypsin Inhibitor Activity (mg/g)	
	RM	ME	RM	ME	RM	ME
SS′B-V1com	752.80 ± 0.02 ^a^	556.6 ± 0.03 ^b^	0.00 ± 0.00 ^a^	0.00 ± 0.00 ^b^	210.60 ± 14.60 ^a^	169.53 ± 0.93 ^b^
CSB Plus	884.54 ± 4.60	---------	0.00 ± 0.00	----------	55.39 ± 1.01	-----------

SS′B = sorghum-soy blend; com = blend that had raw materials procured commercially; CSB = corn- soy blend; RM = raw material; ME = milled extrudate; CE = catechin equivalent. Values in the same row not sharing the same superscript are significantly different at *p* < 0.05.

## Data Availability

The original contributions presented in the study are included in the article, further inquiries can be directed to the corresponding author.

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
