# Peer review of "Characterization of Extruded Sorghum-Soy Blends to Develop Pre-Cooked and Nutritionally Dense Fortified Blended Foods"

_foods, 2025, doi:10.3390/foods14050779_

Round 1

Reviewer 1 Report

Comments and Suggestions for Authors

This study investigates the development of fortified blended foods (FBFs) using sorghum-soy blends (SSB) as an alternative to genetically modified (GMO) grains. The extrusion process improved starch gelatinization, enhanced digestibility, and reduced anti-nutritional factors, such as phytic acid and trypsin inhibitors. The resulting FBFs met established quality standards for energy, protein, and micronutrient content, showing their potential as a viable, nutritious option for food aid programs.

Abstract
Please provide a clear definition of GMO (genetically modified organisms) to ensure clarity for all readers.

Introduction
This section provides sufficient background and includes relevant references. However, some information seems unnecessary and shifts from a scientific to a politically-oriented perspective. To maintain an academic tone, I recommend removing or minimizing content related to U.S. food aid programs. Additionally, the recommendations regarding improving nutritional targets could be better placed in the discussion section.

Materials and Methods
The extrusion processing section lacks an experimental design. The conditions (temperature profile, feed rate, exit die diameter) and sample specifications (moisture content, formulation) appear to be selected arbitrarily. Why were these specific conditions chosen? Was there prior optimization?

There are writing errors in the extrusion processing (2.2) and phytic acid (2.13) sections. Please clarify the drying process equipment and conditions in section 2.2.

For the DSC section, include references that explain how starch gelatinization and total cooking are calculated.

Results and Discussion
There are no references cited in this section.

Table 2 should be referenced in the text before being inserted into the manuscript. Also, indicate significant differences with superscript letters rather than subscript. The letters in the last two columns should have a consistent format.

Adjust the scale of all figures to reduce excessive blank space.

Please review the writing in section 3.5. Additionally, the change in viscosity profile due to a 20°C variation warrants further discussion.

While the first two paragraphs of section 3.8 provide interesting information, they do not directly contribute to the discussion of the research results. I suggest either referencing related reports or removing this section altogether.

Conclusions
The suggestion to publish future results from other binary blends does not belong in the conclusions section. This should be reserved for a summary of the current study’s findings.

Overall Comments
The manuscript has some minor writing mistakes that need to be addressed for clarity and consistency.

Author Response

Translator        

Please, find attached a point-by-point response to the suggestions, comments and corrections from Reviewer 1.

Reviewer 2 Report

Comments and Suggestions for Authors

Dear Author,

I have reviewed your manuscript, Characterization of Extruded Sorghum Soy Blends to Develop Precooked and Nutritionally Dense Fortified Blended Foods. Below are some suggested revisions to improve the quality, clarity, and overall scientific rigor of your paper.

General Comments:

The manuscript contains grammatical errors, awkward phrasing, and inconsistent terminology. A thorough revision for clarity, conciseness, and professionalism is recommended.

In the Abstract, please provide the full form of "GMO" upon first mention and use the abbreviation thereafter.

Specific Areas for Improvement:

Introduction:

Ensure clear problem statement and justification for the study.

Materials and Methods:

Specify missing methodological details where necessary (e.g., precise shear rates in viscosity analysis).

Results and Discussion:

1. Data Referencing Errors:

The text contains multiple instances of “Error! Reference source not found.” in key data sections, including Lines 404, 479, 486, and 567. Please verify and correct these missing citations.

2. Missing Enzyme Activity Validation:

Trypsin inhibitor reduction is reported, but enzymatic activity validation is not provided. Please include relevant data or discussion on its functional significance.

3. Undefined Energy Density Comparison:

Energy density is compared to previous fortified blended foods (FBFs), but specific references or comparative values are not cited. Please provide supporting references.

4. Error in Digestible Starch Increase:

The claim of a 149.65% increase in rapidly digestible starch (RDS) lacks clarity regarding the baseline and final values. Please specify these values.

5. Methodological Inconsistency in Viscosity Analysis:

The pasting property discussion does not specify shear rates or reference comparable studies. Consider adding these details for a stronger methodological foundation.

6. Incomplete Statistical Reporting:

Some results lack p-values or standard deviations, making it unclear whether differences are statistically significant. Ensure statistical details are provided consistently.

7. Ambiguous Reduction in Anti-Nutritional Factors:

The statement that tannins were “not found” post-extrusion does not clarify whether they were initially present in measurable quantities. Please specify initial levels or detection limits.

Addressing these points will significantly enhance the quality and impact of your paper. Please let me know if you require any further clarification.

Best regards,

Comments on the Quality of English Language

Dear Author, 

The manuscript presents valuable research; however, the quality of English requires significant improvement for clarity, professionalism, and readability. 

  • A professional proofreading and editing session is strongly recommended to enhance clarity and readability.
  • If possible, using a native English-speaking editor or a language editing service will improve fluency.
  • Consider breaking down complex sentences and removing redundant phrases for a more engaging and professional scientific tone.

Author Response

Translator        

Please, find attached a point-by-point response to the suggestions, comments and corrections from Reviewer 2.

Reviewer 3 Report

Comments and Suggestions for Authors

This manuscript gives interesting results of the investigation in which the soybean sorghum mixtures were extruded and nutritionally fortified for purpose of obtaining instant porridge products. The characterization of each tested sample was done in details. The topic of this manuscript may be interesting not just for people from academia that deals with grain extrusion and food technology but also for professional from the food industry. Also, this paper may be a good starting point for further investigations on further product improvement. Overall, the layout of the paper is sufficiently good as well as the English language. The used methodology is clear and detailed but it could be further improved, especially in the processing part. The main problem with the manuscript is that due to some formatting or software error, there is lack of many references in the section of text in which results and discussion are presented, which hinders understanding of the current discussion. This should be improved in order to review this work in a complete manner. My other comments are:

Line 57: Define this abbreviation when it is first time present in the text. Abstract should be seen as a separated section.

Lines 60-77: This part seems redundant.

Line 104: What is the optimum of protein and energy? What are the levels considered as optimum?

Line 139: What were nutritional targets? Did you made your own targets based on previous research or there are available values form food industry some kind of document?

Line 167: Please give more info on extrusion process, particularly screw speed and screw configuration.

Line 173: Which device was used for drying?

Line 176: How the cooling of extrudates was achieved?

Line 230: Is oil added slowly by pouring it in mixture during mixing or it was added completely in the beginning of mixing? Or perhaps some type of nozzle was used for oil addition?

Author Response

Translator        

Please, find attached a point-by-point response to the suggestions, comments and corrections from Reviewer 3.

Round 2

Reviewer 1 Report

Comments and Suggestions for Authors

Materials and Methods

Extrusion Processing

The extrusion processing conditions are stated to be based on multiple prior runs to optimize them. However, there is no experimental design or optimization provided. What type of experimental design was used? What were the factors and responses? What criteria were employed for the optimization? This information must be included.

All equations should be numbered and cited in the text.

DSC

Regarding the total cook percent, the lack of a reference could be justified with a proper explanation. However, the following argument: "The total cook percent was shown because for a food to be considered cooked, every component, especially starch and protein, must be cooked," is not satisfactory, as there is no information indicating protein denaturation or starch gelatinization. A more acceptable and referenced parameter could be cooking loss.

Results and Discussion

The phrase "Error! Reference source not found." continues to appear in this section and must be corrected.

Table 2 is still included in the manuscript without being cited prior to its appearance.

Figure 4 needs to be revised as it appears to be overlapped.

Conclusions

The suggestion to publish future results from other binary blends does not belong in the conclusions section. Instead, you could mention what future studies might explore, but this should be framed from a prospective viewpoint.

Author Response

Please, see the attached document. We appreciate the new set of comments to help improve our paper.

The error message seen by the reviewer is definitely not from our end. We assume this might be from the submission. We did not see the same error message when we downloaded our submitted documents in Word format. Perhaps, the reviewer may want to choose to download our response in Word document instead of PDF. 

Reviewer 3 Report

Comments and Suggestions for Authors

-

Author Response

We appreciate the time investment of the reviewer and for their useful feedback in the first round of review. Additional corrections were made to the re-submitted version of our manuscript.